Sargasso Sea bacterioplankton community structure and drivers of variance as revealed by DNA metabarcoding analysis

Gill John Geoffrey 1
Hill-Spanik Kristina M. 1
Whittaker Kerry A. 2 3
Jones Martin L. 4
Plante Craig 1 plantec@cofc.edu
1 Grice Marine Laboratory, College of Charleston , Charleston, SC , United States
2 Sea Education Association , Woods Hole, MA , United States
3 Maine Maritime Academy , Castine, Maine , United States
4 Department of Mathematics, College of Charleston , Charleston, SC , United States
Kormas Konstantinos
Electronic publication date: 2022 Feb 28
Publication date: 2022
Volume: 10
Electronic Location ID: e12835
Received 2021 Aug 17; Accepted 2022 Jan 4
Copyright: © 2022 Gill et al.
Copyright year: 2022
Copyright holder: Gill et al.
License: This is an open access article distributed under the terms of the Creative Commons Attribution License, which permits unrestricted use, distribution, reproduction and adaptation in any medium and for any purpose provided that it is properly attributed. For attribution, the original author(s), title, publication source (PeerJ) and either DOI or URL of the article must be cited.
License URL: https://creativecommons.org/licenses/by/4.0/

Keywords: Biogeography, 16S rRNA, Neutral modelling, Marine microbial diversity

Funding: College of Charleston This work was supported by a small research grant from the College of Charleston. The funders had no role in study design, data collection and analysis, decision to publish, or preparation of the manuscript.

==============================
Marine microbes provide the backbone for pelagic ecosystems by cycling and fixing nutrients and establishing the base of food webs. Microbial communities are often assumed to be highly connected and genetically mixed, with localized environmental filters driving minor changes in structure. Our study applied high-throughput Illumina 16S ribosomal RNA gene amplicon sequencing on whole-community bacterial samples to characterize geographic, environmental, and stochastic drivers of community diversity. DNA was extracted from seawater collected from the surface (N = 18) and at depth just below the deep chlorophyll-a maximum (DCM mean depth = 115.4 m; N = 22) in the Sargasso Sea and adjacent oceanographic regions. Discrete bacterioplankton assemblages were observed at varying depths in the North Sargasso Sea, with a signal for distance-decay of bacterioplankton community similarity found only in surface waters. Bacterial communities from different oceanic regions could be distinguished statistically but exhibited a low magnitude of divergence. Redundancy analysis identified temperature as the key environmental variable correlated with community structuring. The effect of dispersal limitation was weak, while variation partitioning and neutral community modeling demonstrated stochastic processes influencing the communities. This study advances understanding of microbial biogeography in the pelagic ocean and highlights the use of high-throughput sequencing methods in studying microbial community structure.

Introduction

Marine bacteria are amongst the most abundant organisms in the global ocean and are responsible for greater than half of primary productivity and nutrient fluxes in marine environments (Azam et al., 1983; Azam, 1998). The field of microbial biogeography has expanded due to the advancement of genetic tools capable of classifying microbial taxa on a whole-community scale (Giovannoni & Stingl, 2005; Franzosa et al., 2015; Santoferrara et al., 2016). This has allowed investigation into the dispersal capacity of these planktonic microbes, determining the extents to which dispersal barriers, environmental filters, and stochastic effects structure their communities over space and time (Schwob et al., 2021; Vergin et al., 2017). By analyzing the whole community at once, high-throughput genetic sequencing tools can help to resolve the mechanisms driving these sweeping patterns in diversity.

Environmental filters influencing marine bacterial community structure may include physical factors such as temperature and surface mixing depth, chemical variables including salinity or nutrient concentrations, and biological parameters and interactions (Follows & Dutkiewicz, 2011; Friedline et al., 2012; Fuhrman et al., 2008). Marine bacteria are known for their ability to rapidly respond to changing environmental conditions (Corno et al., 2007), allowing for community structures to swiftly change to suit present conditions.

The hypothesis of cosmopolitan distributions, or that “everything is everywhere, but the environment selects,” has remained appealing for decades for its conciseness, but is impossible to fully test given its near-infinite criteria for the influences of time and space with respect to microbial diversity (Baas Becking, 1934). As genetic tools have advanced to directly probe this theory, distance effects and geographic provincialism have been identified as contributing factors to microbial community dissimilarities on global scales, suggesting that barriers to and means of dispersal can have an impact on microbial community structuring (Hörstmann et al., 2021; Logares et al., 2018; Martiny et al., 2006). Physical features, such as defined pycnoclines or ocean currents delineating water masses, may restrict the free, passive dispersal of microbes, bringing about these community dissimilarities over certain time scales (Fuhrman, Cram & Needham, 2015; Hellweger, van Sebille & Fredrick, 2014; Hörstmann et al., 2021; Treusch et al., 2009). Many taxa of marine bacteria are found globally, but local distributions of endemic microbes are frequently identified, particularly in extreme environments (Giovannoni & Stingl, 2005; Pedrós-Alió, 2006).

Stochastic influences may also play a significant role in structuring communities. A neutral community model, based upon dynamics of birth, death, and immigration, has been successfully applied to prokaryotic assemblages (Sloan et al., 2006), demonstrating that chance events play a significant role despite microbes’ rapid responses to deterministic environmental factors and competition (Giovannoni & Stingl, 2005). These tend to influence community composition as a historical contingency, with random fluctuation in mortality or natality (ecological drift), selection for past environmental conditions, or barriers to interbreeding between distant or physically separated populations leading to dynamics of community structuring in the present (Hanson et al., 2012; Sloan et al., 2006). Stochasticity and environmental determinism are thought to co-influence microbial community structure (Caruso et al., 2011). For instance, extremely low-resource environments have been shown to exert more severe limitations on community competition, thus increasing the role of determinism (Stegen et al., 2012), but neutral models have been successfully applied even in resource-poor cryptic desert bacterial communities (Caruso et al., 2011).

The Sargasso Sea, a distinctly bounded open water mass just offshore of the eastern United States, presents an appropriate ecosystem to study mechanisms structuring bacterioplankton communities. High sea surface temperatures and a regional geostrophic downwelling zone contribute to vertical stratification of the water column and oligotrophic conditions, providing annually consistent physical structuring of the water column (Steinberg et al., 2001). These conditions select for picoplankton such as marine bacteria as a productive base, as these communities are able to more effectively absorb nutrients compared to the larger eukaryotic phytoplankton dominant in higher-nutrient areas (Cotti-Rausch et al., 2016; Finkel et al., 2010; Herlemann et al., 2011). Currents of the North Atlantic gyre, including the North Equatorial Current, Gulf Stream, and Antilles Current, provide natural biogeographic boundaries for the Sargasso ecosystem; these gyre currents also congregate the floating holopelagic Sargassum that imparts ecological uniqueness to the Sea (Ardron et al., 2011). The eddies distributed throughout the Sargasso provide additional biogeographic features, where a region of higher eddy influence from the Gulf Stream may be distinguished from downwelling of the central gyre, although the ecological implications of eddy influence are not fully understood (Ardron et al., 2011). The physical dynamics of the Sargasso Sea have been largely explored with regard to Sargassum-associated species (e.g., Govindarajan et al., 2019; Sehein et al., 2014; Siuda, 2011), and the physical, chemical, and biological makeup of the area has been well documented in the Bermuda Atlantic Time Series survey, which has surveyed the same region of the northern Sargasso Sea monthly since 1988 (Lomas et al., 2013).

Surveys of bacterial communities in the Sargasso Sea and broader North Atlantic Ocean have identified depth and chlorophyll-a (chl-a) concentration as significant determinants of community structuring (Cotti-Rausch et al., 2016; Sjöstedt et al., 2014). Other studies focusing on bacterial distribution have found distinct communities at the surface, the deep chlorophyll maximum (DCM), and the upper mesopelagic, as well as distinct communities during periods of winter convective mixing compared to other seasons (Treusch et al., 2009). Water bodies, both on a meso-scale with eddies and surface mixing, as well as on a Longhurstian provincial scale, have also been found to influence prokaryote β-diversity (Cotti-Rausch et al., 2016; Hahnke et al., 2013).

This study sought to further investigate the geographic, environmental, and stochastic factors that influence bacterial communities in the Sargasso Sea and neighboring regions of the North Atlantic using Illumina high-throughput amplicon DNA sequencing. We focused our efforts on characterizing bacterial communities both vertically within the euphotic zone and horizontally across environmentally-defined biogeographic regions. We hypothesized that there would be no geographic dispersal limitation for marine bacteria within the euphotic zone, extending from the surface to the DCM. Of the contemporary environmental filters examined, we expected sampling depth to be the most significant influence on community structure, with distinct communities at the surface and at the DCM driven by distinct nutrient and light characteristics in these two regions of the water column. We further anticipated that stochastic effects would significantly impact distribution and occurrence of these marine bacteria, despite extreme environmental filtering, given their small size and the lack of dispersal barriers in the open ocean.

Materials and Methods

Sampling

Water samples (N = 40) were collected during the daytime from the Sargasso Sea and adjacent areas of the Atlantic Ocean during mornings from 21 April to 18 May 2018. The SSV Corwith Cramer (Sea Education Association; (SEA)) followed a route travelling from Nassau, Bahamas to Pennos Bay, Bermuda to New York City, New York, USA (Fig. 1) as a part of the SEA’s annual transect through the Sargasso Sea (Govindarajan et al., 2019; Law et al., 2010; Sehein et al., 2014; Zettler, Mincer & Amaral-Zettler, 2013). Cruise plans were filed with the US State Department, who obtained the required collection permits. No permits were required for sampling in international and US waters under federal jurisdiction. The cruise and permit numbers for the samples collected in this study are as follows: C279, US State Department Cruise F2017-112, Bermuda permit number SP171103 and Bahamas permit number MAMR/FIS/13.

Figure 1 Map of research cruise track and stations.

Oceanographic regions delineated by white bars and sampling stations indicated by white circles. Darkest gray = land masses. White to gray background gradient indicates relative depth from shallow to deep. NA, North Atlantic; NSS, North Sargasso Sea; SSS, South Sargasso Sea; B, Bermuda; NC, North Carolina, USA; SC, South Carolina, USA. Figure generated using Ocean Data View 4.7.8 (Schlitzer, 2016).

A shipboard flow-through system continuously monitored sea surface temperature (SST), salinity, and turbidity. SST measurements were used to define oceanographic regions observed throughout the research cruise and refine the location of the subtropical convergence zone (STCZ). Neuston tows conducted at each scientific station were used to collect surface Sargassum and zooplankton, and surface Sargassum abundance was weighed manually. Water samples were collected by bucket manually (surface only) or collected using a Seabird Electronics SBE19PlusV2 Conductivity, Temperature, Depth (CTD) Carousel (Bellevue, WA) equipped with a Model QSP-2300 photosynthetic active radiation (PAR) sensor, Sea Point in-vivo chl-a fluorometer, and twelve 2.5-L Niskin bottles. Water samples were collected both from surface waters and from the DCM as determined from the chl-a maximum detected by the CTD fluorometer. Samples from six of 21 scientific stations were filtered in situ using a McLane Research Laboratories Large Volume Water Transfer System (Falmouth, MA, USA). Due to equipment malfunctions, remaining samples for molecular analysis were collected by vacuum-filtering 250–500 mL of water through a 0.22-µm Millipore mixed-cellulose membrane filter. All filtered microbial samples were stored at −20 °C on board. Both filtration methodologies (in-situ McLane filtration and Niskin bottle-collected shipboard filtration) involved vacuum filtration of seawater onto mixed-cellulose membrane filters with identical pore sizes (0.22 µm), and downstream sample processing, from filter handling to extraction to sequencing, was identical across samples. In-situ sampling (from the McLane pump) may have been associated with a lower probability for bacterial contamination, given the demand for less seawater handling. However, we bioinformatically examined evidence for bacterial contamination across our samples (detailed below), allowing us to confidently compare samples collected in situ with those collected via Niskin bottle/CTD and filtered on board.

Nitrate concentrations corresponding with each sample were determined via cadmium reduction and colorimetric methods, measured using an Ocean Optics Flame spectrophotometer (Orange County, FL, USA). To determine chl-a concentration, seawater was first filtered over 0.45-µm GF/F filters. Chl a was then extracted by immersing filters in 5 ml of 90% acetone, and extractions were performed in the dark at −20 °C. Chl-a concentrations were determined using a Turner Designs Model 10-AU Benchtop Fluorometer (San Jose, CA, USA) following Strickland & Parsons (1972).

Oceanographic regions

Oceanographic regions used for statistical analysis and categorization of the data were defined by geographic and oceanographic factors. The South Sargasso Sea (SSS) and North Sargasso Sea (NSS) boundary, typically set in the literature at 29°N (Sjöstedt et al., 2014; Siuda, 2011) was modified for this study to fall between samples taken at 27.2°N and 28.5°N (Fig. 1) based on a decrease in SST at this latitude (SST < 24 °C), indicating the location of the STCZ during the study. The Antilles Current (AC) and Sargasso Sea boundary was established using an RDI Ocean Surveyor 75 kHz Acoustic Doppler Current Profiler (Teledyne RD Instruments, Poway, CA, USA) as well as at-depth CTD data indicating the presence of a warm (>26 °C) water mass near the DCM. The North Atlantic (NA) was defined as the region beyond the influence of the Gulf Stream along the continental shelf of the northeastern US, as measured using the RDI current profiler. Sampling was also conducted offshore of Bermuda during a port stop. The Bermuda sample is included in descriptive data summaries for comparison to the open-ocean sites, but was excluded from all statistical analyses due to small sample size.

DNA extraction and library preparation

DNA was extracted from the 0.22-µm water filters using a QIAGEN DNeasy Plant Mini Kit (Germantown, MD) following the manufacturer’s protocol. DNA was amplified following a modified version of the 16S Metagenomic Sequencing Library Preparation protocol using S-D-Bact-0341-b-S-17/S-D-Bact-0785-a-A-21 primers (Klindworth et al., 2013), targeting hypervariable regions 3 and 4 of the 16S ribosomal RNA (rRNA) gene identified for use in next-generation sequencing-based diversity studies, which included forward and reverse Illumina overhang adapters, respectively. Triplicate 25-µl reactions were performed using 1x TaKaRa Ex Taq Buffer (Clontech, Mountain View, CA, USA), 0.4 µg ml−1 bovine serum albumin, 0.3 mM dNTPs, 0.2 µM of each primer, 0.025 U µl−1 TaKaRa Ex Taq DNA polymerase (Clontech), and 1 µl template DNA. Cycling was as follows: 95 °C for 3 min; 30 cycles at 95 °C for 30 s, 50 °C for 30 s, and 72 °C for 30 s; a 5 min final extension at 72 °C. Samples were electrophoresed on 1% agarose gels stained with GelRed (Biotium, Hayward, CA, USA), and triplicate products were pooled based on band intensity to approximately equalize concentrations in the pool. All PCR negative controls were also pooled and were processed along with samples to identify any potential contamination during library preparation. All pooled products were indexed using a unique combination of forward and reverse Illumina Nextera indices, and indexed PCR products were cleaned using AMPureXP beads (Beckman Coulter, Brea, CA, USA) following the manufacturer’s protocol. Successful indexing was confirmed using gel electrophoresis as above, with a subset being analyzed on a Bioanalyzer 2100 using a DNA 1000 kit (Agilent Technologies, Santa Clara, CA, USA). Cleaned, indexed products were quantified using the dsDNA Broad Range assay for an Invitrogen Qubit 1.0 fluorometer (Thermo Fisher, Waltham, MA, USA) and then pooled in equimolar ratios based on the sample with the lowest concentration. The pooled library was quantified using a Qubit 1.0 (Thermo Fisher, Waltham, MA, USA) and ran on the Bioanalyzer 2100 (Agilent) as above. The library was sequenced on an Illumina MiSeq (San Diego, CA, USA) using a 2 × 250 bp sequencing kit at the Medical University of South Carolina Hollings Cancer Center Genomics Shared Resource (Charleston, SC, USA).

Sequence data analysis

Sequencing data were demultiplexed by the sequencing facility. Sequence quality was examined using FastQC (www.bioinformatics.babraham.ac.uk/projects/), and primers and adapter sequences were trimmed from each sequence using Cut-adapt (Martin, 2011) in the Galaxy platform (galaxyproject.org). Fastq-join (Aronesty, 2013) in Quantitative Insights into Microbial Ecology (QIIME) v. 1.9.1 (Caporaso et al., 2010) was used to join forward and reverse reads with 8% maximum difference between matching segments and a 6-nucleotide minimum length difference. QIIME was also used to remove sequences with quality scores <24. USEARCH V.5.2.236 (Edgar, 2017) was then used to detect and remove chimeras using UCHIME (Edgar et al., 2011) and cluster operational taxonomic units (OTUs) at a 97% sequence similarity threshold. A representative set of sequences was selected and then compared to the Greengenes 16S rRNA database (DeSantis et al., 2011) using the RDP classifier to assign taxonomy to each representative sequence (Wang et al., 2007). Any OTU found in the negative control was considered a contaminant if the number of reads from any sample was <10X the number of reads from the negative control (as in Pagenkopp Lohan et al., 2017). Based on these criteria, three contaminant OTUs identified as g. Moraxella, Streptococcus, and Micrococcus were removed from the dataset.

A single rarefaction was performed to equalize the number of OTUs to the sample with the fewest number of sequences (N = 15,468) prior to statistical analyses. Shannon diversity, number of observed OTUs, Heip’s evenness (E), and Chao1 α-diversity indices were calculated for each sample using QIIME and used to determine species richness and evenness. For phylogenetic-based β-diversity analysis (weighted UniFrac; Hamady, Lozupone & Knight, 2009), sequences were aligned in QIIME using PyNAST (Python Nearest Neighbor Alignment Space Termination, Caporaso et al., 2009). The alignment was filtered to remove positions where a gap was present in 80% of the sequences. The top 10% most entropic base positions were also removed. An approximately maximum-likelihood phylogenetic tree was created using FastTree 2.1.3 (Price, Dehal & Arkin, 2010) and used to create a weighted UniFrac dissimilarity matrix (Hamady, Lozupone & Knight, 2009).

α-diversity metrics between depths and among oceanographic regions were compared using Kruskal-Wallis tests in JMP (SAS, Cary, NC, USA). The effect of environmental and geographic variables on α-diversity was explored using multiple linear regression. Standard least square models with eight (date, latitude, longitude, temperature, salinity, nitrate, chl a, Sargassum density) or nine environmental variables (adding depth) were run after a square-transformation on α-diversity metrics (observed OTUs, Chao1, Shannon, and evenness) due to lack of normality with raw data. β-diversity was compared across categories (oceanographic region or depth) using PERMANOVA tests in QIIME, with PERMDISP tests conducted in R version 3.6.3 (R Core Team, 2015) using the betadisper function in vegan (Oksanen et al., 2020) to assess sampling dispersions. Pairwise PERMANOVA tests were conducted in QIIME on significant results to assess the specific communities accounting for statistical separability. Principal coordinates analysis (PCoA) was used to visualize community associations and results were plotted using EMPeror (Vázquez-Baeza et al., 2013) in QIIME. Two-way SIMPER (similarity percentage) analysis, performed in PRIMER-E v6 (Clarke & Gorley, 2006), was used to identify taxa responsible for differences between geographic regions and depths. Venn diagrams were generated in VENN (http://bioinformatics.psb.ugent.be/webtools/Venn/) to examine the number of shared and unique OTUs among sampling locations at the surface and at the DCM.

Partial distance-based redundancy analysis (db-RDA) was applied to examine relationships between bacterioplankton community structure (weighted UniFrac distances) and environmental variables (date, temperature, salinity, nitrate, chl a, depth, Sargassum density), while controlling for geographic distance. Using the capscale function in vegan (Oksanen et al., 2020), forward selection was used to determine the subset of environmental factors that best explained community variation, as determined by adjusted R2 values. The principal coordinates of neighbor matrix (PCNM; Borcard & Legendre, 2002) eigenfunctions, which represent the spectral decomposition of the spatial relationship across sampling locations, were considered as the spatial variables in the ordination-based analyses. Variation partitioning was carried out to partition community variation into environmental and spatial effects, enabling us to determine the variation in the bacterial community explained by the various unique and combined fractions of spatial and environmental data. Because the shared fraction of variation was negative, we used proportional apportioning (Legendre, Borcard & Roberts, 2012) to correct the variation attributable to each unique factor.

We also evaluated the fit of a neutral community model (NCM) to determine the potential contributions of stochastic processes on bacterial community assembly. NCMs were fitted with least squares to mean relative abundance and detection frequency data (Keymer, Lam & Boehm, 2009) for each sampling site, based on the equations originally derived by Sloan et al. (2006). Sloan’s model is an adaptation of the neutral theory (Hubbell, 2001), specifically tailored to deal with extremely large population sizes and the inability to completely describe microbial species abundances. For these analyses, we separately employed the surface, deep, and combined datasets. R2 values represent goodness of fit to the NCM, and larger R2 values indicate stochastic dispersal and/or ecological drifts are more important than selection to community assembly (Logares et al., 2013; Sloan et al., 2006). All computations were performed in R version 3.6.3 (R Core Team, 2015).

Results

Environmental parameters

DCM depth varied throughout the sampling track from ~40 to 150 m (Table 1). At the surface, temperature and salinity ranged from 14.8 to 25.7 °C and from 33.7 to 36.9 psu, resp.; at depth, values ranged from 19.5 to 26.4 °C and from 36.4 to 36.9 psu, resp. (Table 1). Temperature exhibited a significant negative correlation with latitude (r = −0.88, P < 0.001). Nitrate concentrations were below detection level for all surface samples except the single sample from Bermuda (0.69 µM), which was taken in an urbanized harbor, and one outlier in the SSS (0.15 µM) (Table 1). Nitrate in the DCM ranged between 0 and 1.57 µM. Sargassum density varied from 0.03 to 0.35 g m−1 (Table 1).

Table 1 Mean environmental values by depth and region.

Depth category	Region	N	Depth (m)	Temp (°C)	Salinity (psu)	Nitrate (µM)	Chl-a (µg g−1)	Sarg. dens. (g m−1)	
Surface	B	1	–	ND	ND	0.69	0.283	0	
	NA	2	–	18.5 (3.75)	35.1 (1.38)	BD	0.144 (0.09)	0.03 (0.03)	
	NSS	6	–	22.9 (0.24)	36.1 (0.36)	BD	0.030 (0.01)	0.03 (0.02)	
	SSS	9	–	25.3 (0.10)	36.7 (0.04)	0.69 (0.02)	0.023 (0.00)	0.12 (0.04)	
DCM	AC	3	146.6 (10.2)	26.3 (0.10)	36.5 (0.06)	0.02 (0.02)	0.232 (0.08)	–	
	NA	2	42.5 (22.5)	16.7 (2.85)	36.1 (0.59)	0.09 (ND)	0.126 (0.07)	–	
	NSS	5	94.0 (8.3)	20.1 (0.24)	36.8 (0.03)	0.64 (0.11)	0.273 (0.07)	–	
	SSS	12	87.8 (11.8)	23.3 (0.55)	36.7 (0.09)	0.36 (0.10)	0.214 (0.03)	–	
Note:

Columns describe: depth category (DCM, deep chlorophyll maximum), oceanographic region (AC, Antilles Current; NA, North Atlantic; NSS, North Sargasso Sea; SSS, South Sargasso Sea; B, Bermuda), number of samples taken (N), depth, temperature (Temp), salinity, nitrate concentration, chlorophyll-a (chl-a) concentration, Sargassum density (Sarg. dens.). No samples were taken at depth in Bermuda due to gear deployment restrictions at port, and no samples were taken at the surface in the Antilles Current due to modifications in sample collection. ND indicates samples for which contamination or equipment maintenance prevented collection of data, BD indicates samples for which particular concentrations were below detectable levels.

α-level biodiversity

An average of 1621 OTUs (±65) per sample was found in the 39 open-ocean seawater samples. Overall, the DCM samples demonstrated significantly greater species richness than surface samples as indicated by the higher number of observed OTUs (X2 = 5.46, df = 1.00, P = 0.020) and Chao1 index (X2 = 4.51, df = 1.00, P = 0.034) (Table 2 and Table S1). This distinction was attributable to significant differences in observed OTUs, Chao1 index, and Shannon biodiversity between DCM and surface samples in the SSS (Table S1). Small sample sizes prevented similar regionally-specific analysis of the two depth categories elsewhere.

Table 2 Mean α-diversity values by depth and region.

Depth category	Region	N	H′	S	E	Chao1	
Surface	B	1	7.47	1,787	0.10	3,214	
	NA	2	7.24 (0.14)	1,157 (48.0)	0.13 (0.01)	2,122 (62.8)	
	NSS	6	8.13 (0.07)	2,150 (13.9)	0.13 (0.01)	4,940 (72.3)	
	SSS	9	7.86 (0.13)	1,710 (17.0)	0.14 (0.02)	3,409 (80.7)	
DCM	AC	3	8.21 (0.23)	1,707 (47.9)	0.17 (0.02)	3,878 (266.8)	
	NA	2	7.70 (0.14)	1,754 (42.5)	0.12 (0.00)	3,708 (306.5)	
	NSS	5	8.21 (0.11)	2,183 (9.22)	0.14 (0.02)	4,894 (81.5)	
	SSS	12	7.54 (0.18)	1,715 (21.6)	0.11 (0.02)	3,732 (82.8)	
Note:

Columns describe: depth category (DCM, deep chlorophyll maximum), oceanographic region (AC, Antilles Current, NA, North Atlantic, NSS, North Sargasso Sea, SSS, South Sargasso Sea, B, Bermuda), number of samples taken (N), Shannon biodiversity index (H′), number of observed OTUs (S), Heip’s evenness (E), and Chao1 richness index. Values in parentheses indicate standard error.

α-diversity did not significantly differ among oceanographic regions whether sample sets were considered as a whole or subdivided by comparable depths (Table 2, Table S1). No environmental variables had significant predictive ability in regression models for any of the four diversity metrics (P > 0.05).

Community composition

Proteobacteria was the most commonly observed phylum in all regions, both in surface waters and at the DCM (~55–69%), followed by Cyanobacteria (9–17%) and Bacteroidetes (6–11%). The Bermuda sample was most distinct, with notably higher contribution of the Proteobacteria families Rhodospirillaceae and Rhodobacteraceae, and much higher dominance of Synechococcaceae relative to Prochloraceae (Fig. 2). This single surface sample from Bermuda coastal waters was more dominated by abundant OTUs than communities in pelagic environments, as indicated by its low evenness (E = 0.10; Table 2).

Figure 2 Community structure of all identified operational taxonomic units (OTUs) making up >3% of all samples taken from (A) surface waters and (B) the deep chlorophyll maximum (DCM).

Relative abundances do not sum to 100% as less abundant taxa were not included to allow for clear visualization of the data. Sequences of OTUs were compared to the National Center for Biotechnology Institute’s GenBank database using BLAST (basic local alignment search tool; Altschul et al., 1990) to determine taxa identities to the lowest taxonomic level possible. All sequences from this dataset were 98–100% similar to sequences from GenBank, but most GenBank sequences were not identified to species level and only to genus, family (f), order (o), or class (c). Note that N = 1 for Bermuda and this sample was obtained from surface waters (A), and all Antilles Current samples (N = 3) were taken at the DCM (B). NA, North Atlantic; NSS, North Sargasso Sea; SSS, South Sargasso Sea; AC, Antilles Current; B, Bermuda.

Within the DCM, sequences from Pelagibacteraceae and Alteromonadaceae (Proteobacteria), and Alphaproteobacteria OTU 1 were dominant in all regions, but sequences from OTU 1 were relatively more dominant in the South Sargasso Sea (Fig. 2). As in the surface communities, Synechococcaceae and Prochloraceae (Cyanobacteria) sequences were abundant in the DCM. In the AC and both NSS and SSS samples, sequences from Prochloraceae were more common than those from Synechococcaceae, whereas the reverse was true in the NA samples.

Community dissimilarity ranged from 62–74% based on two-way SIMPER analysis, with the NSS and the AC being the least dissimilar (Table S2). No single OTU contributed more than 0.88% to the dissimilarity among communities among regions at each sampling depth, with 49 to 64 OTUs contributing to 10% dissimilarity between regions at each depth (Table S2).

The number of unique OTUs at a given sampling location at each depth exceeded that of shared OTUs among all sampling locations (805 OTUs for surface, 737 for DCM samples), with the exception of NA surface waters (Fig. 3). Overall, 71.5% of OTUs identified in surface samples were unique to their region, while 52.7% of OTUs identified at the DCM were unique to their respective regions. A total of 37.3% of all OTUs observed appeared in only a single sample.

Figure 3 Relationship between number of unique and shared operational taxonomic units found among (A) surface sampling locations and (B) deep chlorophyll maximum sampling locations.

The diagram was generated using VENN (http://bioinformatics.psb.ugent.be/webtools/Venn/). NSS, North Sargasso Sea; SSS, South Sargasso Sea; NA, North Atlantic; AC, Antilles Current.

Distance-decay of bacterial community similarity

We observed no relationship between bacterial community similarity and geographic distance among all samples analyzed together (R2 = 0.001, F1,1480 = 2.49, P = 0.115; Fig. 4A). When analyzed separately, the surface samples exhibited a significant but weak decay of similarity with geographic distance (R2 = 0.037, F1,270 = 11.43, P < 0.001). No distance-decay relationship was observed among DCM samples (R2 = 0.001, F1,460 = 1.58, P = 0.210; Fig. 4).

Figure 4 Distance-decay relationship between bacterial community similarity (weighted UniFrac) and geographic distance between sampling sites.

(A) All data combined, (B) surface samples only, and (C) deep chlorophyll maximum samples only.

β-diversity and community structure

Community structures were significantly, but weakly, separable based on oceanographic region (N = 39, pseudo-F = 1.892, P = 0.028; Table S3), but not by depth (N = 39, pseudo-F = 0.492, P = 0.832). When DCM and surface communities were considered independently and compared by oceanographic region, greater separability was observed in each, with surface communities being more separable by region (N = 17, pseudo-F = 4.640, P = 0.001) compared to the DCM (N = 22, pseudo-F = 2.765, P = 0.003). Pairwise comparisons between oceanographic regions indicated that differences between microbial communities in the SSS and NSS regions primarily accounted for these results (N = 15, pseudo-F = 8.252, P = 0.002 at the surface; N = 17, pseudo-F = 4.300, P = 0.004 at the DCM) (Table S3). PERMDISP results revealed significant differences between sample group dispersion based on oceanographic region (F-value = 5.65, P = 0.003), but not based on sampling depth. Comparisons between depths at each oceanographic region revealed a significant difference between NSS surface and DCM communities (N = 11, pseudo-F = 14.73, P = 0.004) (Table S3).

PCoA analysis conducted on all collected samples revealed overlap among communities across the geographic range of the study but demonstrated a clear separation of surface and DCM communities collected in the NSS. Surprisingly, the most obvious overlap in communities was NSS surface and SSS DCM bacterial communities (Fig. 5).

Figure 5 Principal coordinates analysis plot based on weighted UniFrac distances comparing bacterial communities from the oceanographic regions sampled.

AC, Antilles Current; SSS, South Sargasso Sea; NSS, North Sargasso Sea; NA, North Atlantic. Solid plot points represent communities from the deep chlorophyll maximum, outlined points represent surface communities.

Spatial and environmental determinants of microbial community structure

Partial db-RDA performed with all environmental variables (temperature, salinity, chl a, nitrate, depth, Sargassum density, and date) indicated that that a three-variable model including temperature, nitrate concentration, and Sargassum density gave the best fit after partialling out spatial effects (F3,17 = 2.24, P = 0.014), although only temperature and Sargassum density were significant individual factors (F1,17 = 2.38, P = 0.042 and F1,17 = 2.75, P = 0.024, resp.). No models were significant (P > 0.05) when redundancy analysis was performed separately for surface or DCM samples. Variation partitioning resulted in a negative shared fraction of variation (i.e., environmental and spatial influence), resulting from nonorthogonality of a key environmental variable and the spatial factor—temperature and geographic distances were significantly correlated (R2 = 0.443, P = 0.001; Mantel test). After correcting with proportional appropriation, environmental and spatial variables could together explain ~28% of the variation in bacterial community structure. Pure spatial effects explained approximately ~11% of the variation, while pure environmental factors explained 17%. Total residual variation was high at 72%.

The NCM explained a large fraction (R2 = 0.86) of the variability in occurrence frequency of the entire bacterioplankton metacommunity of the Sargasso Sea and surrounding North Atlantic (Fig. 6). When surface and DCM samples were analyzed separately, the NCM likewise explained a large proportion of variability (R2 = 0.84 and 0.83, resp.; Fig. 6).

Figure 6 Fit of the neutral model of community assembly.

(A) All, (B) surface-only, and (C) deep-only bacterioplankton. Red lines indicate the best fit to the model as determined in Sloan et al. (2006). R2 indicates the fit to the neutral model.

Discussion

Due to its clear biogeographic boundaries and internal patterns of environmental structuring, the Sargasso Sea and adjacent waters provided an ideal setting to study the drivers of bacterioplankton community structure. We employed high-throughput DNA sequencing to compare communities from surface waters and the DCM, as well as among oceanographic regions. Overall, we found a weak pattern of distance-decay and relatively small differences in community structure among regions, although β-diversity was somewhat higher for surface waters than the DCM. Environmental selection, especially temperature, was a stronger driver of community variation than dispersal limitation. We also found that stochastic factors contributed significantly to biogeographic patterns observed.

Bacterial species richness was greater in the DCM than at the surface, with this difference being most pronounced in the South Sargasso Sea. Previous studies have found that bacterial richness increases with depth through the water column up to the 0.1% light level (Treusch et al., 2009), peaking near the bottom of the epipelagic and gradually declining with increased depth (Wang et al., 2020). The trend of increasing diversity with depth is often amplified in oceanic regions such as the Sargasso Sea with significant stratification and oligotrophic surface conditions (Gajigan et al., 2018; West et al., 2016), where surface waters are heavily impacted by rainfall, sun exposure, flotsam, and other conditions that may permit more dynamic fluctuations of opportunistic bacterial taxa.

Bacterioplankton composition was in general agreement with prior studies conducted in epipelagic waters of the open ocean. We found that sequences from phyla Proteobacteria, Cyanobacteria, and Bacteroidetes, in descending order of relative abundance, comprised the bulk of bacterioplankton, which corroborates earlier studies from the epipelagic zone (e.g., Pommier et al., 2007; Sjöstedt et al., 2014; Zheng, Dai & Huang, 2016). The predominance of sequences from the SAR11 clade was unsurprising because it is widely thought to be the most abundant planktonic microbe in the world’s oceans (Giovannoni, 2017; Morris et al., 2002). Observations of sequences and relative abundances from the cyanobacteria Prochlorococcus and Synechococcus are also in accord with prior research. For instance, Sjöstedt et al. (2014) observed these cyanobacteria in the Sargasso Sea in high relative abundance and, as in our study, noted that Prochlorococcus sequences were more prevalent than those from Synechococcus. In contrast, in the North Atlantic, we observed a relative dominance of Synechococcus sequences over those from Prochlorococcus, consistent with previous studies and likely due to cooler water temperatures (Flombaum et al., 2013). Despite the broad similarity in dominant bacterial groups among all regions and both depths, communities were distinct among the various sampling locations, either due to differences in relative abundances or presence of less abundant OTUs.

Neither species richness nor biodiversity was found to be related to latitude or regional water masses, contrasting with some studies that have suggested that these community characteristics may be shaped by decreasing temperatures at higher latitudes or water-mass specific dispersal barriers and oceanographic differences (Agogué et al., 2011; Costello & Chaudhary, 2017; Moran, 2015). Although the question of a latitudinal diversity gradient for bacterioplankton has been addressed by numerous investigators, no clear answer has emerged. For instance, Sul et al. (2013) reported a negative correlation between latitude and bacterioplankton diversity, whereas others have reported non-linear trends, with diversity maxima at intermediate latitudes (Fuhrman et al., 2008; Raes et al., 2011; Sunagawa et al., 2015). Trends with latitude have been attributed to temperature relations (Sunagawa et al., 2015), while other studies have found stronger relationships with primary productivity or other environmental factors (e.g., Raes et al., 2011).

Regional differences in β-diversity were statistically significant, but weak, and the PCoA plot showed regional overlap, highlighting the relative influences of both environmental filters and neutral processes in shaping bacterioplankton communities. When surface and DCM communities were analyzed separately across all regions, surface communities demonstrated greater separation than deep communities, particularly within the Sargasso Sea itself. This disparity was corroborated by the slightly stronger distance-decay relationship for our surface samples as compared to the DCM. Communities from all regions were not statistically separable based on sampling depth. However, there was a clear separation of surface and DCM communities when examining only those samples from the North Sargasso Sea. The distinct structuring and biodiversity characteristics support our hypotheses that environmental pressures differing at the surface and the DCM would lead to distinctions in these two microbial communities. Given unbalanced sampling design and significant PERMDISP results, however, the differences in these community structures (as per PERMANOVA) could be attributed to any combination of differences in group location centroid or sampling dispersion. The PCoA plot demonstrating general clustering of samples based on their region and depth (i.e., surface Sargasso Sea samples were separated from their respective DCM samples) provides qualitative evidence that differences in community structure were in fact due to differences in location centroids but we cannot conclusively rule out difference due to statistical dispersion.

Prior studies likewise observed provincialism among bacterial assemblages of the upper pelagic (Friedline et al., 2012; Gómez-Pereira et al., 2010; Milici et al., 2016), and our results also concur with several studies that found no distance-decay for bacterioplankton within the DCM (Walsh et al., 2015; Milici et al., 2016). This pattern emphasizes the importance of contemporary environmental factors in community assembly, as surface bacterioplankton are likely subjected to more variable physical-chemical conditions than those in deeper waters (Ghiglione et al., 2012).

Variation partitioning revealed that combined environmental factors accounted for approximately 17% of the variation among bacterial assemblages. Independently, the spatial component explained 11% of the dissimilarity among communities. The distance effect indicates that processes other than contemporary selective factors have left a legacy on current microbial community compositions, e.g., through past selection or ecological drift. At least some level of dispersal limitation is required for past events to leave a contemporary signature, otherwise community changes due to past selection or drift would be counteracted. Therefore, a significant distance effect can be interpreted as evidence of dispersal limitation over some time period (Hellweger, van Sebille & Fredrick, 2014). Significant influence from both environmental and spatial factors has been identified in 68% of studies reporting on the subject, according to a literature review by Hanson et al. (2012).

Environmental selection, primarily temperature, was stronger than dispersal limitation in accounting for variation among bacterial assemblages (17% vs 11%); this finding agrees with several prior studies of marine bacterioplankton (Hanson et al., 2012; Hu et al., 2020; Milici et al., 2016; Sunagawa et al., 2015). Relatively weaker spatial patterning in bacteria supports the “size-dispersal” hypothesis, i.e., that smaller organisms have higher dispersal rates, especially when dispersal is primarily passive (De Bie et al., 2012). In contrast, some investigators have noted a stronger influence of dispersal limitation relative to environmental factors in bacterioplankton (Wu et al., 2018), in support of the ‘size-plasticity’ hypothesis, i.e., that smaller organisms are less environment-filtered compared to larger organisms because they are more likely to be plastic in metabolic abilities and have greater environmental tolerances. Several recent studies have likewise examined the biogeographic patterns of microeukaryotes, at times in direct comparison to bacteria, to shed additional light on assembly processes of marine microbes. With respect to the size-plasticity vs size-dispersal hypotheses, findings of these comparative studies have been mixed—some have supported the size-dispersal hypothesis (Logares et al., 2018, 2020), while others supported the size-plasticity hypothesis (Wu et al., 2018; Allen et al., 2020), and still others that showed little difference between bacteria and protists with respect to relative roles of spatial and environmental effects (Bock et al., 2020; Milici et al., 2016).

Redundancy analysis in this study demonstrated the importance of temperature in determining bacterioplankton community structure. In addition, the clustering of South Sargasso DCM and North Sargasso surface samples in the PCoA analysis showed that although these two water masses are separated by distance, as well as light level, salinity, and other abiotic factors, their respective communities were more similar to one another than to those from the same depth category in other regions, likely due to their temperature similarity (Table 1, Fig. 5). Further, the only region with a significant difference in communities with respect to depth (the North Sargasso) also had the greatest difference in temperature (2.8 °C) compared to differences between depths at other regions. One study focusing on widely distributed marine microbes such as SAR11 and Actinobacteria, which we found to be dominant in Sargasso Sea communities, indicated that temperature significantly influences community dissimilarity, even when environmental differences associated with distance are eliminated (West et al., 2016). Further, seasonal mixing events and productivity cycles created by shifts in temperature regimes have also been shown to impact bacterial diversity (Wang et al., 2020).

Sargassum density was also identified as an important environmental influence on bacterial assemblages based on our spatially-controlled redundancy analysis. While Sargassum and other flotsam and debris found in the Sargasso Sea have been shown to harbor distinct and diverse microbial communities (Zettler, Mincer & Amaral-Zettler, 2013), the buoyant seaweed’s absence from DCM environments equivocates its importance on correlated variables, including temperature or unmeasured variables such as light availability, which may instead drive differentiation with depth.

A large fraction (72%) of variation among bacterial assemblages in our variation partitioning analysis was left unexplained. Hanson et al. (2012) found that on average 50% of variation in microbial composition was unexplained, although that fraction ranged widely among studies. Unexplained variation is typically attributed to unmeasured environmental factors or biotic interactions (Vass et al., 2020), but may also be due to stochastic processes (Hanson et al., 2012; Plante, Fleer & Jones, 2016).

The NCM explained a large portion (~86%) of the variation in occurrence frequency of photic-zone bacteria in the Northwest Atlantic Ocean, including for separate analyses of surface waters and the DCM. Our findings are in general accord with Mo et al. (2018) who likewise found stochastic factors to be primary in bacterioplankton community assembly in open-ocean habitat, albeit at a somewhat lower level of dominance (57%). However, several other studies found deterministic factors to be more important relative to neutral processes in similar scenarios (Allen et al., 2020; Hu et al., 2020; Vergin et al., 2017; Wu et al., 2018). Although spatial factors comprise a portion of the neutral process (Hubbell, 2001), the strength of the NCM in our study does not contradict the relatively small spatial effect revealed by variation partitioning. Spatial factors (i.e., dispersal limitation) comprise just one part of the neutral process, whereas other stochastic processes such as ecological drift also contribute to community dissimilarity (Sloan et al., 2006, 2007). The significant but weak distance-decay pattern that we observed at the surface also indicates a low level of dispersal limitation (Hanson et al., 2012). The large stochastic effect is likely ascribed to high and random dispersal of bacteria from a common, regional source pool (Chen et al., 2017).

Conclusions

In this study, we compared bacterial community structures at the surface and at the DCM within the Sargasso Sea and adjacent oceanographic regions. Community structures were separable on the basis of physically-defined oceanographic regions and separable by depth in the North Sargasso Sea. While both environmental differences and physical distance were found to be correlated with greater community dissimilarity, these dissimilarities were amplified in surface communities, and stochastic effects explained a significant proportion of variability in bacterial occurrence frequency, demonstrating the complexity of interactions governing this community. Future research is needed to further investigate biological interactions or co-occurrences, which could help drive community structuring, and to understand the role of a wider range of physical and chemical variables, such as dissolved oxygen content or particulate matter.

Supplemental Information

Supplemental Information 1 Kruskal-Wallis analysis of variance tests comparing α-diversity statistics from bacterial communities.

Categorized (A) by categorical sampling depth (surface vs deep chlorophyll maximum (DCM)); and (B) by oceanographic region. Significant values (α < 0.05) are indicated in bold. H′ = Shannon diversity, S = number of observed operational taxonomic units, E = Heip’s evenness, SSS = South Sargasso Sea, NSS = North Sargasso Sea.

Click here for additional data file.

Supplemental Information 2 Rank-order correlations (Spearman’s ρ) between environmental variables and α-diversity metrics of bacterial communities from the Sargasso Sea and surrounding areas of the Atlantic.

P-values are indicated in parentheses. S = observed operational taxonomic units, H′ = Shannon diversity, E = evenness.

Click here for additional data file.

Supplemental Information 3 ANOSIM (analysis of similarity) results comparing bacterial communities at differing depths and in differing oceanographic regions.

(A) communities throughout water column compared based on depth and based on region, (B) communities from one portion of the water column compared by region, (C) communities from within the Sargasso Sea compared by region and by depth. SSS = South Sargasso Sea, NSS = North Sargasso Sea.

Click here for additional data file.

We would like to offer a special thanks to the crew, researchers, and students aboard SEA Semester’s Marine Biodiversity and Conservation program, cruise C-279, for making our work at sea possible. This is Grice Marine Laboratory contribution number 576.

Additional Information and Declarations

Competing Interests

Author Contributions

Field Study Permissions

Data Availability

The authors declare that they have no competing interests.

John Geoffrey Gill conceived and designed the experiments, performed the experiments, analyzed the data, prepared figures and/or tables, authored or reviewed drafts of the paper, and approved the final draft.

Kristina M. Hill-Spanik analyzed the data, prepared figures and/or tables, authored or reviewed drafts of the paper, and approved the final draft.

Kerry A. Whittaker conceived and designed the experiments, authored or reviewed drafts of the paper, and approved the final draft.

Martin L. Jones analyzed the data, prepared figures and/or tables, authored or reviewed drafts of the paper, and approved the final draft.

Craig Plante analyzed the data, authored or reviewed drafts of the paper, and approved the final draft.

The following information was supplied relating to field study approvals (i.e., approving body and any reference numbers):

Cruise plans were filed with the US State Department, who obtained the required collection permits. No permits were required for sampling in international and US waters under federal jurisdiction.

The following information was supplied regarding data availability:

The raw sequencing data and associated metadata are available at the Sequence Read Archive database (Leinonen, Sugawara & Shumway, 2010): PRJNA753104.

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
