# Peer review of "Sargasso Sea bacterioplankton community structure and drivers of variance as revealed by DNA metabarcoding analysis"

_PeerJ, doi:10.7717/peerj.12835_

## Round 0.1 · original submission · Major Revisions

Please provide a detailed point-by-point rebuttal letter to each of the reviewers' comments, along with your revised manuscript.

Reviewer 1 ·

Basic reporting

peerJ review
Review of paper: “Biogeographic analysis of bacteria in the Sargasso Sea using
high-throughput sequencing”. The paper presents some interesting findings from the Sargasso Sea regarding the drivers of bacterial community assembly. The authors applied a range of data analysis methods to investigate the relative role of environmental selection over stochasticity in shaping community assembly. The overall aim of the study is interesting, however I have some concerns with the amount and structure of the data available as well as the suitability of data analysis methods to support the questions set and the findings of the study. Due to insufficient amount of information in the methods, I’m not able to assess at this point whether the manuscript could be publishable. I would expect the concerns raised below to be addressed before further consideration.

Lines 60-62 & Lines 77-79. There are plenty of more recent studies on the topic applying novel methodologies from both bacteria and protists. Reading here seems a bit limited.
Lines 94-103: Need to explain why this area is distinct by contrasting it with adjacent seas. Are there barriers that make dispersal limited…eg ocean currents, or some geomorphological features?
122-128: The hypothesis is unclear as the authors are not comparing the dispersal limitation processes within the euphotic zone with a different condition, eg in the disphotic zone. Also the information presented in lines 94-103 seem rather disconnected from the hypothesis whereas the hypothesis should rather build on the info previously provided. In short, it is unclear what the authors are hypothesising and why.
127-128: In my mind, a community is either assembled based on a dominant force or by stochastic processes, however, this last phrase seems to imply that it can be both, which I find confusing. The reasoning behind this last paragraph needs to be much more thought out and connected to background presented higher up.
132-134: The “sampling” section should provide adequate details on the areas sampled, stations within areas. The map should provide also a map of the broader geographical area of the easter states inside which this smaller frame should be indicated. …”Sargasso Sea and adjacent areas”..you need to explain what are the similarities differences between these Seas and the limits of these areas on the map and why they were selected -connect your original research questions and hypotheses with your experimental design. The total number of samples n=40 needs to be broken down to the detailed experimental design that it was derived from so the readers can understand what samples were collected from where and why (eg 2 areas x 10 sites x 2 depths? = 40). Also provide more detail on the legend of figure 1: eg what does the gray gradient color stand for? I guess depth, but you need to be precise.
146-148: You need to be upfront about the bias of using different methodologies can pose on your findings (if any).

151-153: you need to state according to which protocol of analysis and add appropriate reference.

155-167: Ok now I understand why the areas were not defined in the first paragraph of the methods. You used the env info collected to categorise the stations within different areas a posteriori. In which case I am wondering what was the reasoning behind this transect design in the first place? You will need to better connect your hypothesis with the reasoning behind the way you did your sampling.
I will leave the details of the molecular analysis to someone more expert on this topic.
223: You need to state which a-diversity metrics you used and what they do/show, justify why you selected the specific ones and add appropriate references.
224-225: It is unclear why the authors chose to do a correlation analysis instead of fitting a statistical model with the diversity metric as the response variable and the env variables as explanatory also adding depth and area as factor variables. The model selection procedure would help them test the effect of all these geographical and env variables on the diversity simultaneously. This cannot be done with correlation analysis as such I don’t think this analysis is appropriate here.
226-229: you will need to explain what the analysis does and how it helps you address your research question. Eg what do you mean compared across categories, what are the categories in your case, need to put this into the context of your own study.
231-233: Again it is unclear how this analysis helps address the research questions. In the Mantel test, geographical distances are included as covariates not accounting for the fact that two stations that are very close can actually belong to two areas with widely different characteristics. This analysis does not thus take into account dispersal limitations between the sites.
236-237: You use Venn to determine the overlap between which things and why? You need to be more specific in your description of methods and how these connect to your objectives.
237-242: This analysis seems much more suitable to understand the role of environmental variables in driving community similarity as the role of geographical distance and its overlap with env variables is accounted for. Thus to me the mantel tests seems rather obsolete, unless I’m missing something here.
246: Again it is unclear what the question is behind this method and how it relates/complements the rest of the analysis carried out here. As it stands it seems rather disconnected from the rest of the text. You should clearly explain how this approach was employed to address a specific question as well as what the results would look like and how you will use them to address the question.
General remark: the authors need to explain how the unequal number of samples across areas (eg see table 1) is affecting their analysis. In my opinion this unequal sampling design and the fact that in some areas some variables were not measured, would have an important bias in the validity of the results.
268-273: Unsure how chi squared analysis was used to test for significant differences between depths here…Again a GLM with consequent post hoc tests seems more suitable.

329: Are you able to say which regions are significantly different from which?
Tables 1 and 2 need to say what numbers represent, eg mean, median and add some measure of variation eg range
343-344: This sentence is very confusing. Not sure how temperature differences can be correlated with something.
349-352: Why doesn’t this model include area? Is it because it can only include numerical variables instead of factors? The structure of this model should have been described in the methods.

357: What is meant here by spatial effects? Even though spatial effect had a smaller contribution than env drivers, it would be useful to visualise in which axes did most of the effect occur so as to be able to unravel the mechanism. Eg northeast or from south to north?

Experimental design

See above

Validity of the findings

Discussion
First paragraph of the discussion: Please first state your main findings and how these address your research questions and hypotheses.

391-392: these conclusions are unsupported by the results as a GLM analysis is needed instead of correlation.

410-413: this aim should be stated also clearly in the aims and objectives paragraph of the intro…

It would be useful to see some discussion whereby the relative role of spatial structures and the environment is contrasted between bacteria and protists. There have been several interesting studies on the later that can provide further insights here.

Reviewer 2 ·

Basic reporting

The manuscript agrees with PeerJ’s requirements for basic reporting. The study presents a biogeographic analysis of microbial communities in the Sargasso Sea and adjacent regions and aims to identify the environmental, spatial and stochastic factors that influence the observed distribution. The raw sequencing data and associated metadata were deposited into the Sequence Read Archive database however these will be released on 30/9/2022. Overall, the study has merit although there is space for improvement on structure, clarity and conciseness.

Experimental design

The research lies well within the aims and scope of the journal. The research question is well-defined but the authors could state more clearly how this research fills an identified knowledge gap given the fact that the microbial communities of the Sargasso Sea are among the best-studied worldwide. Concerning the bioinformatic analysis, I would expect a more up-to-date sequence clustering method resulting in ASVs rather OTUs however, I will not insist on that. What is more important is to make the manuscript more reader-friendly and I am referring mainly to the statistical analysis. Reading the article, the impression I got was that every single statistical test available has been used without really having thought of the question that it will answer. Several times, the same or similar question, was addressed by a number of statistical methods resulting in a confusing mix. Also, insignificant results are presented when there is no need for that. The authors should think carefully which methods or tests they are using and why. Lastly, there is one sample that is completely different from the rest, the coastal Bermuda sample. It would be best to remove this sample altogether, it cannot be used in any stats and including it resulted in unnecessarily complex tables of statistical results with or without this sample.

Validity of the findings

The findings are valid but some restructuring and reduction in length is needed. I would recommend presenting the “big picture” first, i.e., the beta-diversity analysis, then go into the detail of the factors and taxa that may be important in the observed differences. The core finding is that microbial communities differ only slightly and, on this basis, the authors can then explore, the variation partitioning between environmental and spatial or stochastic factors that cause these subtle differences.

Additional comments

Line 94-96: Please expand on why the Sargasso Sea, which is a distinctly bounded open water mass, is an appropriate ecosystem to study biogeographic processes. I presume due to the subtropical convergence zone?
Line 113: The last paragraph of the introduction is a good place to address the novelty aspects of this study compared to previous ones (see comment in #2 Experimental design). What new does this research bring? Also, the word “vertically” is somewhat misleading, one expects a vertical profile of the whole water column but it’s just two depths that are considered here.
Lines 117-121: This part spoils the flow of the text here. It would fit better in the Materials and Methods.
Line 124-128: It is better to rephrase this part and refer to all the environmental parameters tested (including stochastic effects) to confirm your hypothesis, rather than an expectation that one of them is significant among the tested factors. This is something to be tested in the analysis later.
Figure 1: It would help to show the wider area in a smaller map on the side of the main one.
Line 149: typo in colorimetric
Line 173: Please mention the region of the 16S rRNA gene that your primers are targeting.
Line 209-210: please reconsider this sentence to improve clarity
Line 214: Do you mean prior to statistical analysis? Wasn’t the rarified dataset used for all downstream statistics?
Line 216: Did you use Unifrac or weighed Unifrac?
Table 1. Please remove the line ALL, it is not informative provide an average of T or chlorophyll of all regions.
Lines 267-294: Please use more concise language. Here for example, the take-home message is that there was a statistically significant difference between surface and DCM attributed to the SSS samples. This could be conveyed in one sentence. Also please use the same number of decimals throughout the manuscript for the same metric, e.g., 3 decimal places for all p values.
Line 285: Is it not Bacteroidetes?
Line 286-288: Please remove any mention to Archaea if your primers were not targeting them. This was part of the filtering process and should not be in the Results.
Line 283: This is where it starts becoming confusing. According to SIMPER we expect these communities to be quite dissimilar but then ANOSIM and PCoA give a different impression.
Line 329: Again, please be more concise. The whole paragraph should be shortened to convey a single outcome: only surface communities differ between regions. You could then perform pairwise testing to see where the significant difference stems from. You can do that with Permanova and pairwise permanova. I am not sure what “overlapping but separable” means from an ecological point of view.
Line 338: About Mantel tests: since you have high correlation between two variables and you need to use partial Mantel test controlling for that, then only this result should be presented. It becomes more confusing further on with db-RDA (I presume on the same distance matrix as Mantel) and other factors become significant. Also, it is not clear to me how you ended up with the best fit model. What kind of selection process was used and were there any interaction terms? It is mentioned that 3 terms are in the best fit model but only two are significant. Is that because of an interaction term? If this is the case then only the significant interaction is mentioned and not the individual variables that make up this interaction.
Line 359: Why is the residual variance so high when 58% of the variation is constrained?
Lines 367-378: In the first part of the paragraph, it is mentioned that diversity peaks at a certain point and then declines which agrees with your data. I think the statement below about deep water stable conditions etc. cannot be supported by your data since you don’t have any deep water samples. The paper you are referring to (Ghiglione et al. 2012) is comparing surface to deep ocean (>200 m depth) phylogenetic diversity and speciation patterns. In this study, all samples belong to the surface ocean. I think you should focus on the special characteristics of DCM rather than depth-related changes in diversity.
Line 431: if there is no reasonable oceanographic reason that these two are similar, I would avoid these statements. Other samples are close temperature-wise but appear to be dissimilar.
Line 444-446: this sentence is unclear, please explain better. Also consider if Sargassum density is really significant or a statistical artifact with no ecological meaning.
Line 453-455: This should be the core of the discussion i.e., the acknowledgement that this is a well-bounded water mass and differences in microbial communities are subtle. On this basis, you can then explore, the variation partitioning between environmental and spatial or stochastic factors.
Line 459: Is there an agreed dissimilarity level over which the contribution of legacy factors are considered important? How about the partial Mantel test that shows no effect of geographic distance when controlled for temperature? This is what I meant before about becoming lost in the statistics, finding significant differences and factors causing dissimilarity, to then reach the conclusion that is mentioned in the previous comment.
Figure 4: I am not familiar with distance-decay models but it seems that there are points having a similarity of 1 and 0 distance. I presume it refers to one community being compared to itself, if that makes any sense. Shouldn’t these be excluded from the regression?

---

## Round 0.2 · accepted · Accept

I thank the authors for their detailed and targeted revision to all of the reviewers' comments, including re-analysing part of their data. The provided revision renders the paper acceptable for publication.